# Peridomestic natural breeding sites of *Nyssomyia whitmani* (Antunes and Coutinho) in an endemic area of tegumentary leishmaniasis in northeastern Argentina

**Mariana Manteca-Acosta** [1,2]*, **Regino Cavia** [2,3], **María Eugenia Utgés** [1], **Oscar Daniel Salomón** [2,4], **María Soledad Santini** [1,2]

**1** Centro Nacional de Diagnóstico e Investigación en Endemo-epidemias, ANLIS-Malbrán, Ministerio de Salud de la Nación, Buenos Aires, Argentina, **2** Comité Nacional de Investigaciones Científicas y Técnicas (CONICET), Buenos Aires, Argentina, **3** Departamento de Ecología, Genética y Evolución, Facultad de Ciencias Exactas y Naturales, Universidad de Buenos Aires e Instituto de Ecología, Genética y Evolución, Buenos Aires, Argentina, **4** Instituto Nacional de Medicina Tropical, ANLIS-Malbrán, Ministerio de Salud de la Nación, Misiones, Argentina

* mariana.manteca@gmail.com

**Data Availability Statement:** All relevant data are within the manuscript.

## Abstract

The scarce information about breeding sites of phlebotomines limits our understanding of the epidemiology of tegumentary leishmaniasis. Identifying the breeding sites and seasons of immature stages of these vectors is essential to propose prevention and control strategies different from those targeting the adult stage. Here we identified the rural breeding environments of immature stages of *Ny. whitmani*, vector species of *Leishmania braziliensis* in the north of Misiones province, Argentina; then we determined and compared the environmental and structural characteristics of those sites. We also identified the season of greatest emergence and its relationship with adult abundance. During a first collection period, between 28 and 48 emergence traps were set continuously for 16 months in six environments of the farm peridomicile and domicile: below house, chicken shed, experimental chicken shed, forest edge, pigsty and under fruit tree. Traps were checked and rotated every 40 nights. A total of 146 newly emerged individuals were collected (93.8% of them were *Ny. whitmani*), totaling an effort of 23,040 emergence trap-nights. The most productive environments were chicken shed and below house, and the greatest emergence was recorded in spring and summer. During a second collection period, emergence traps and light traps for adult capture were placed in the chicken shed and below house environments of eight farms. Emergence traps were active continuously during spring, summer, and early autumn. Environmental and structural characteristics of each environment were recorded. A total of 84 newly emerged phlebotomines (92.9% *Ny. whitmani;* 72,144 emergence trap-nights) and 13,993 adult phlebotomines (147 light trap-nights) were recorded in the chicken shed and below house environments. A positive correlation was also observed between trap success of newly emerged phlebotomines and of adults after 120 days. A high spatial variability was observed in the emergence of *Ny. whitmani*, with the number of newly emerged individuals being highest in soils of chicken sheds with the highest number of chickens and closest to

**Funding:** The authors received no specific funding for this work.

**Competing interests:** The authors have declared that no competing interests exist.

forest edge. Moreover, below house was found to be as important as chicken sheds as breeding sites of *Ny. whitmani*. Management of the number of chickens in sheds, soil moisture and pH, and the decision of where to localize the chicken sheds in relation to the houses and the forest edge, might contribute to reduce the risk of human vector exposure and transmission of *Leishmania*.

## Author summary

Phlebotomines are the subject of much research because of the role of their females as the only proven natural vectors of *Leishmania* species, the parasitic protozoans that are the causative agents of the neglected tropical disease leishmaniasis. The lack of ecology knowledge of immature stages of this vector limits their prevention and control strategies. We identified the rural breeding environments of immature stages of *Ny. whitmani*, that is the vector of tegumentary leishmaniasis in the north of Misiones province, Argentina, and compared the environmental and structural characteristics of those sites. We also identified the season of greatest emergence and its relationship with adult abundance. These results open the possibility of experimental evaluation of environmental management practices for prevention and control of the immature stages through strategies different from the ones usually conducted for adults.

## Introduction

Phlebotomines are dipteran insects (Psychodidae: Phlebotominae) of sanitary importance; they transmit species of protozoan flagellates of the genus *Leishmania*, causal agents of leishmaniases [1]. These diseases are a health problem worldwide, including many regions of the Americas [2]. In Argentina, tegumentary leishmaniasis (TL) is an endemic disease, with epidemic outbreaks in the subtropical northern region. *Leishmania braziliensis* (Vianna) (Kinetoplastida: Trypanosomatidae) is the main etiological agent of the disease. In northwestern Misiones province, in the region of the triple border with Brazil and Paraguay, the phlebotomine species *Nyssomyia whitmani* (Antunes and Coutinho) is the main vector [3].

The most common TL transmission scenario is the sylvatic environment, with people working in the native forest being at risk. However, in rural, ruralized periurban environments, and in the urban-rural interface, where forest patches are spatially contiguous, peridomestic transmission has also been reported [4–6], which may be due to recent adaptations of the vector species. Indeed, the last outbreak recorded in the area surrounding the locality of Puerto Iguazú (Misiones) occurred in a rural area known as "Dos Mil Hectáreas" in 2004; there, the phlebotomine species *Ny. whitmani* was found to be naturally infected with the parasite [7]. This species was also the most prevalent in the outbreak, particularly in animal shelters close to recently deforested patches [7–9].

Different types of breeding sites of phlebotomine species have been described in the world. They are usually humid sites rich in organic matter, such as soil in the tropical forest [10,11], soil of animal shelters [12,13], rodent burrows [14], tree roots and trunks [11,15], leaf litter [16], termite colonies [14], and caves and rock crevices [17]. However, the low collection frequency and low abundance of captured individuals indicates that only a few of these breeding sites can be regarded as stable [18].

In the present century, while research efforts have been focused on the identification of immature stages of vector species, it has not been possible to obtain a number of immature individuals proportional to the high density of adult phlebotomines recorded in the environments [15,19]. Pre-imaginal stages of *Ny. whitmani* were recorded in few works conducted in their distribution area, with no more than five specimens per sample being collected [11,20]. Reinhold-castro et al. (2015) collected immature stages of this species in environments near henhouses in Paraná, Brazil, whereas Casanova (2001) found them in the same environments in Sao Paulo, Brazil. In Argentina, only one found immature individuals [21]. After the collection effort, direct observation, and incubation of 27 soil samples, those authors only recorded a single larva of *Migonemyia migonei* (França) from the base of a *Bromelia* sp. and a pupal exuvium in a dog resting site.

Although in the "Dos Mil Hectáreas" area, in Puerto Iguazú, phlebotomine adults were recorded in wild, domestic and peridomestic environments [7,8,22], there is no information about the breeding sites of any of the 17 species described in the area [8], including *Ny. whitmani*. Thus, the aim of this work was to identify natural breeding sites of *Ny. whitmani* in the domestic and peridomestic environments close to the forest edges associated with recent deforestation in an TL endemic area in Argentina, in environments characterized by high transmission risk, and to compare their environmental characteristics, including edaphic ones. Likewise, changes in phlebotomine emergence in those environments over the year were evaluated and variations in the periods of highest emergence were identified and related to adult abundance.

Knowledge of the breeding sites, their environmental requirements, and the annual pattern of *Ny. whitmani*, as well as of other Phlebotominae vectors of *Leishmania* may contribute to the design of integrated management strategies of the vector, incorporating monitoring or targeted control of larval stages, and developing life-cycle adjusted indicators of impact for evaluating the interventions.

## Methods

### Study area

This study was conducted in a rural area known as "Dos Mil Hectáreas" (-25˚ 43' S, -54˚ 35' W), located south of Puerto Iguazú city (-25˚ 36' S, -54˚ 35' W), Misiones, Argentina. The area was originally a native semi-deciduous rainforest described as part of the Paranaense phytogeographic province (Neotropical phytogeographic region, Amazon forest domain) [23]. The climate is subtropical without dry season; mean minimum and maximum temperatures, and relative humidity, are 20˚C 30˚C and 78%, respectively in the warmest month (January), and 10˚C, 22˚C and 83% respectively in the coldest month (July) (taking the 1981–2010 period) [24]. The area has been subjected to deforestation by families that settled there in 2003 (Salomón et al., 2009). Therefore, the farms are usually surrounded by patches of remnant primary and secondary forest; the anthropized domestic area consists of a stilt house built with wood slats 20 cm above the ground, shelters for domestic animals, mainly chicken, pigs, rabbits, and dogs, fruit trees, a cropped area (cassava and/or corn), and an orchard. The study area is also adjacent to the protected areas of Peninsula Provincial Park and near Iguazú National Park.

### Emergence traps

The emergence of phlebotomine imagoes was studied using emergence traps modified from Ferro (1997) and Casanova (2013). The traps consisted of a polyvinyl chloride (PVC, SICA S. A.) pipe tube, 15 cm tall and 10.15 cm in diameter, which covered 0.0081 m2 of ground substrate. The trap was fixed 5 cm into the ground using three steel flat bars (18 cm long) attached

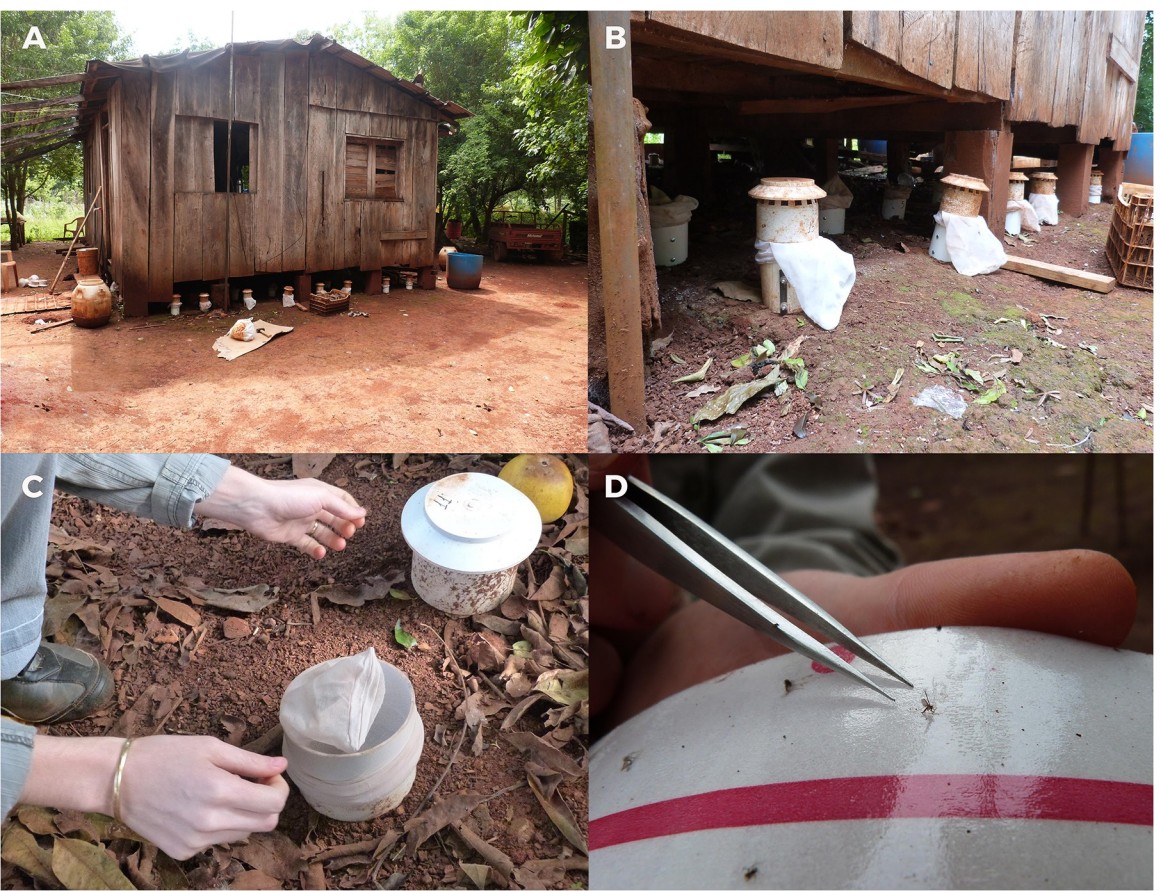

**Fig 1. Emergence traps.** A: Emergence traps fixed under a house. B: Emergent trap fixed to the ground using steel flat bars attached and PVC vent cap attached to each tap to protect. C: Covering the emergence trap with a sand fly-proof fine mesh to prevent sand flies escaping. D: A new emergent adult phlebotomine removed directly of adhesive paper with entomological tweezers.

to the inner surface (Fig 1A and 1C). After fixing the trap, the inner surface of the pipe tube was totally covered with a band (5 cm wide, 30 cm long) of adhesive paper (Fly-catcher DELENTE Eco). To prevent newly emerged phlebotomines from escaping, the top of the trap was covered with voile fabric fixed with a rubber band (Fig 1C). Finally, to protect the trap from the sun, animals and rain, a PVC vent cap was attached to each trap (Fig 1B). The ground of each environment was divided in 40 x 40 cm quadrants and some of them were selected for further sampling. Then, in each selected quadrant, one trap was fixed to the ground. Traps were checked every 15–20 nights to prevent sample deterioration, examined for adhered phlebotomines, and replaced with new adhesive bands. Each phlebotomine captured with these traps was recorded as a new adult that emerged in the period of 15–20 nights between trap revisions as newly emerged individual. Considering a mean period of 40 days for the development of phlebotomine immature stages under laboratory conditions [12], the emergence traps were maintained in each quadrant for 32 to 42 nights (one biological cycle), depending on logistics and meteorological factors. After one cycle, the traps were relocated within the same quadrant, rotating them clockwise approximately 10 cm. Thus, each relocated trap was considered a new emergence trap, as it would catch the newly emerged individuals in an adjacent place in the same quadrant. The newly emerged phlebotomines found in each trap were removed directly with entomological tweezers or by cutting the piece of tape to which it was

attached (Fig 1D). We recorded the total number of newly emerged phlebotomines of each species in each trap and cycle. The capture effort was estimated as the number of trap/nights.

## Sampling design

The study was divided into two collection periods: from March 16[th], 2014 to October 5[th], 2015, and from October 21[st], 2015 to April 6[th], 2016. The first collection period consisted of two phases and the main objectives were to screen possible domestic and peridomestic breeding environments of phlebotomines, and to define the time of the year of greatest emergence. The second period was focused on the breeding environments detected during the first period and aimed to identify the environmental and structural characteristics of the farms, and physico-chemical characteristics of the soil of the sites with a greater number of newly emerged phlebo-tomines, and to study the association between number of newly emerged phlebotomines and abundance of adults.

For the first collection period, a typical farm of the "Dos Mil Hectáreas" area was selected. Phase 1 was conducted from March 16[th], 2014, to March 26[th], 2015 ($\cong$12 biological cycles), in six environments: below house, soil below the elevated house (Fig 1A); chicken shed, soil in the chicken shed; pigsty, bare soil inside the pigsty; under fruit trees, soil immediately around the fruit tree trunk and under the canopy; forest edge, the soil of the farm area contiguous to the edge of the forest; and experimental chicken shed, the soil of the three chicken sheds (with three chicken each) already described in Manteca-Acosta (2017), built on the farm in the area adjacent to the forest edge. As already mentioned, the plot of soil of each environment was divided into 40x40-cm quadrants. According to the operational capacity and quadrant accessibility, 2 to 8 quadrants per plot of ground were selected in each environment. The number of emergence traps varied according to the area available for sampling. Phase 2 was conducted from March 26[th], 2015, to October 7[th], 2015 ($\cong$5 biological cycles) only in the environments where newly emerged phlebotomines were recorded in phase 1 to optimize the search in the farm. In this sampling, 16 quadrants were selected in each environment.

For the second collection period, eight farms were selected in the "Dos Mil Hectáreas" area, including the farm studied in the first collection period. We selected farms of similar area (approximately 5 ha each), close to primary and secondary forest, such as the edge of defor-ested areas, with permanent inhabitants and with chicken breeding in sheds for at least the last four months. Collection was performed from October 21[st], 2015, to April 6[th], 2016 ($\cong$ 4 bio-logical cycles) in the two environments of each selected farm where the greatest abundance of newly emerged phlebotomines was recorded in first collection period. Between 19 and 40 quadrants (occupying 3 to 7 m2) were sampled in each environment, with the number of sam-pled quadrants depending on the number of quadrants where traps were set. Traps were set, checked, and relocated as in the first collection period. In order to study the relationship between the abundance of adults and the number of newly emerged phlebotomines in the environments, sampling of adults was performed using REDILA-BL light traps [25] in both collection periods, at the same time and in the same environments where the emergence sam-pling was conducted. One light trap was placed in each one of two environments of each farm. The light traps were active between one and two consecutive nights per month, coinciding with the dates of the emergence trap rotation. The light traps were installed 1.5 m above the ground from 17:00 h to 8:00 h of the following day. Abundance of adults was estimated by the capture success (number of phlebotomine adults captured by a light trap-nights) (Table 1).

In all cases, each pool of specimens was diaphanized with lactophenol (1:1) to make the spe-cific determination of the specimens of the family Phlebotominae. Subsequently, each phlebo-tomine was placed on a coverslip for sex and species identification under an optical

**Table 1. Environment and structural characteristics recorded on eight farms and physicochemical characteristics of soil taken from three farms of the "Dos Mil Hectáreas" area during the second period of collection of newly emerged phlebotomines: blood sources, structural and spatial characteristics, adult phlebotomine abundance, and soil physicochemical characteristics.**

| Characteristics | Description |
| --- | --- |
| **Abundante of blood sources** | |
| Number of people | Number of people living on each farm |
| Number of dogs | Number of dogs inhabiting and sleeping on each farm |
| Number of chickens | Number of chickens kept in each chicken shed on each farm |
| Density of chickens | Number of chickens per $m^2$ of chicken shed |
| Other animals | Presence or absence of other animals (pigs, rabbits, geese, etc.) on the farms |
| **Structural characteristics** | |
| House elevation | Level of elevation (cm) of the house floor (space under the elevated house) |
| House size | Size ($m^2$) of the house on each farm |
| Chicken shed | Size ($m^2$) of the chicken shed on each farm |
| Resting site | Presence or absence of animals sleeping in the space under the house |
| Management | Presence or absence of management actions (raking, using insecticides, etc.) in each environment |
| **Spatial characteristics** | |
| Distance between environments | Distance (m) between soil plots in each environment on each farm |
| Distance to forest edge | Distance (m) of each soil plot to the nearest primary or secondary forest patch |
| Distance to Provincial Park forest edge | Distance (m) of each plot to the forest edge of the Provincial Park |
| Inverse distance to forest edge | Inverse measure of the variable distance to forest edge |
| Inverse distance to Provincial Park forest edge | Inverse measure of the variable distance to Provincial Park forest edge |
| Proportion of trees | Proportion of trees in a 250-$m^2$ buffer area surrounding each plot |
| **Abundance of phlebotomine adults** | |
| Trap success for adults | Number of phlebotomine adults captured by light trap-nights |
| **Physicochemical characteristics of the soil** | |
| Humidity | Percentage (%) of soil relative humidity |
| pH | Soil pH |
| Total organic matter | Percentage (%) of total soil organic matter |
| Organic matter | Percentage (%) of soil organic matter fractions |
| Organic carbon | Percentage (%) of soil organic carbon fractions |
| Total nitrogen | Percentage (%) of total nitrogen in soil |
| Carbon/nitrogen | Proportion of Carbon/Nitrogen ratio of soil |

microscope, according to Galati (2017) [26]. Species identification was not possible in specimens of *Brumptomyia* sp. (without vectorial competence) and in cryptic species, which were determined to genus level. Since many of the species of the *cortelezzii* complex are cryptic and include females that are not distinguished by morphological traits [27], in the present work these specimens are referred to as *Ev. cortelezzii s.l.*

To identify farms' environmental and structural characteristics that might explain the emergence of phlebotomines, during the second period of collection, we recorded the following data for each farm: number of people and number of different domestic animals as potential blood sources for phlebotomines, structural characteristics of the buildings, spatial characteristics, abundance of phebotomine adults and soil physicochemical characteristics (Table 1). These characteristics were recorded in the field, except for distance from each

environment to the nearest primary or secondary forest patch edge and to the Peninsula Provincial Park forest edge, and proportion of trees, which were estimated using Google Earth. The proportion of the 250-m area surrounding each sampling site occupied by trees was estimated with a regular grid.

Soil physicochemical characteristics (Table 1) were recorded on the farms that showed the highest adult and newly emerged phlebotomine abundance. Soil was collected from both environments on April 21st, 2016, from 8:00 h to 11:00 h, coinciding with the last sampling of emergence traps of the second collection period. A soil sample of 250–300 g was taken by removing the top 5 cm of soil from each quadrant of the studied environment. Samples were taken with a shovel, and leaves, branches and feces were removed; then they were stored at 4˚C to avoid moisture loss during transport to the Cerro Azul laboratory of the Instituto Nacional de Tecnología Agropecuaria (INTA) for analysis [28].

## Data analysis

To characterize the quality of the studied environments as phlebotomine breeding sites (where phlebotomines emerged) in the first and second collection periods, five measures of newly emerged phlebotomines imagoes abundance were considered: number of sampled soil quadrants in each environment where at least one phlebotomine emerged, total number of traps set in each environment where at least one phlebotomine emerged, trap success in each environment (number of newly emerged individuals /number of traps), daily productivity of each environment (number of newly emerged phlebotomines per day and square meter) and total period productivity per environment (number of newly emerged phlebotomines in the total area of each environment on each farm for the total collection period).

Spearman correlation tests with different time lags were used to study the relationship between the abundance of newly emerged and adult phlebotomines. We used emergence trap success in each of the four biological cycles and trap success of adults recorded at different time lags from the same cycle up to 3 cycles time lag (120 nights) of the second collection period [29]. This analysis was performed for each environment separately and by pooling the data from the two environments. The farm environmental and structural characteristics that might explain phlebotomine emergence were subjected to multiple regression analysis using generalized linear mixed models (GLMM). Data from the second collection period were used, following a step forward procedure for the selection of explanatory variables [30]. Given the limited knowledge of the variables involved in the emergence of phlebotomines and the scales at which they are associated, the selected sampling design included evaluation of the characteristics at two levels: (1) at the plot level: the experimental unit was each sampling plot in each environment on each farm (data of all quadrants of a single environment and farm were pooled), and (2) at the quadrant level: the experimental units were the 40x40-cm quadrants in each soil plot. A GLMM with Poisson error structure and logarithmic link function was used to analyze the abundance of newly emerged phlebotomines in each cycle at the plot level, considering the total number of newly emerged individuals captured in the traps per plot and cycle as the response variable, and the number of emergence traps set in each plot and cycle as offset. The farm was considered a random effect, and it was removed when it did not improve model explanation [31]. The environmental and structural variables, as well as the abundance of adults were included in the analysis at this level as explanatory variables. Three estimations of phlebotomine adults were considered: 1- trap success of each environment in the farm; 2- trap success in each farm, averaging all cycles; and 3- trap success in each cycle and farm, by averaging data of both environments. In addition, two fixed variables were incorporated: environment and cycle, the latter representing each one of the four biological cycles included in

the second collection period. For this analysis, data of the eight farms were used; soil physico-chemical characteristics were not included, since they were not available for all farms.

To evaluate possible differences in soil physicochemical characteristics of each quadrant between plots, environments or farms, a redundancy analysis (RDA) was performed using the seven physicochemical soil variables as response variables and the plot, environment and farm as categorical explanatory variables [32]. A backward selection procedure was used to exclude explanatory variables that did not explain the differences in soil physicochemical characteristics. Then, since over-dispersion was observed, a GLMM with negative binomial error structure and logarithmic link function was used to analyze the abundance of newly emerged phlebotomines at the quadrant level. The explanatory variables that were included in this analysis were the same as those included at the plot level as well as the physicochemical characterization of soil recorded in each quadrant (Table 1). The plot and farm were included in the model as random effects, and were removed when they did not improve model explanation [31]. This analysis was conducted using the data from the three farms where soil physicochemical characteristics were recorded.

We used the greatest significant change of deviance for a variable or interaction as the step-wise forward selection criteria, and the simplest significant models were reported for both levels of analysis. We assessed the association between predicted variables using a Pearson correlation test [31] and the variance inflation factors (VIFs); if any of the correlation coefficients was higher than |0.5| or a VIF value was higher than 5, the variable was removed and the process was repeated [31]. When more than one candidate model was found, the Akaike Information Criterion (AIC) was used to select the best models and only the ones with $\Delta AIC < 5$, compared to the best model, were reported [33]. Continuous explanatory variables were standardized and/or rescaled before being entered into the models [31]. The RDA was performed using *vegan* package [34], the GLMMs using *lme4* package [35], and VIFs were calculated with *car* package for R (R Core Team Program, 2017).

## Results

During the first collection period, 146 newly emerged phlebotomines were recorded, with a total effort of 576 emergence trap each one active for 40 nights (23,040 emergence trap-nights): 95 newly emerged phlebotomines captured in phase 1, and 51 individuals in phase 2. In phase 1, newly emerged phlebotomine were recorded in three of six environments: below house (89), chicken shed (5) and pigsty (1). Below house showed the highest values for: number of quadrants with newly emerged phlebotomines, number of traps with newly emerged phlebotomines, trap success and productivity of newly emerged phlebotomines; this environment was followed by chicken shed and pigsty (Table 2). Two species were captured: *Ny. whitmani* (n = 93), *Brumptomyia* sp. (n = 1); it was not possible to identify a male phlebotomine because of the bad condition of the sample. *Nyssomyia whitmani* was found in all three environments, but mostly in below house, whereas *Brumptomyia* sp. was found only in below house. The period of greatest abundance of newly emerged phlebotomines was from October 2014 to January 2015, with a peak in below house during December-January (Fig 2). In phase 2, newly emerged phlebotomines were also recorded in the three sampled environments. Again, below house (44) showed the highest number of quadrants and of traps with newly emerged phlebotomines, and the highest trap success and productivity of newly emerged phlebotomines, followed by pigsty (4) and chicken shed (3) (**Table 2**). Three species were recorded: *Ny. whitmani* (86.3%), *Brumptomyia* sp, (9.8%) and *Mg. migonei* (3.9%). *Nyssomyia whitmani* and *Brumptomyia* sp. were found in all three environments, whereas *Mg. migonei* was found only in below house. Between April and June 2015, all three environments had occurrence of newly emerged phlebotomines (Fig 2).

**Table 2.** *First collection period.* Summary of the newly emerged phlebotomines captured in domestic and peridomestic environments during the first collection period: phase 1 (from March 16th, 2014 to March 26th, 2015–379 nights-12 cycles) and phase 2 (from March 26th, 2015 to October 7th, 2015–195 nights-5 cycles), in a farm located in the "Dos Mil Hectáreas" area, south of Puerto Iguazú city. Size of each environment (Size, m²), Number of quadrants with newly emerged phlebotomines (number of soil quadrants sampled in each environment where at least one phlebotomine emerged), Number of traps with newly emerged phlebotomines (total number of traps set in each environment where at least one phlebotomine emerged), Trap success (number of newly emerged phlebotomines/number of traps), daily productivity of each environment (number of emergent phlebotomine/m²/day) and total period productivity (number of newly emerged phlebotomines in the total area of each environment on each farm for the entire collection period).

| Environments | Size (m²) | N° quadrants with newly emerged phlebotomines | N° traps with newly emerged phlebotomines | Trap success | Daily productivity per square meter | Total period productivity |
|---|---|---|---|---|---|---|
| | | *(N° quadrants) | **(N° traps) | | | |
| *Phase 1* | | | | | | |
| below house | 30 | 6 (8) | 21 (96) | 0.93 | 3.58 | 107.30 |
| chicken shed | 16 | 3 (8) | 4 (96) | 0.05 | 0.20 | 2.41 |
| Exp. chicken shed 1 | 1.5 | 0 (2) | 0 (24) | 0 | 0 | 0 |
| Exp. chicken shed 2 | 1.5 | 0 (2) | 0 (24) | 0 | 0 | 0 |
| Exp. chicken shed 3 | 1.5 | 0 (2) | 0 (24) | 0 | 0 | 0 |
| forest Edge | 1 | 0 (3) | 0 (36) | 0 | 0 | 0 |
| Pigsty | 9 | 1 (2) | 1 (24) | 0.04 | 0.16 | 0.96 |
| under fruit tree | 2 | 0 (1) | 0 (12) | 0 | 0 | 0 |
| **Total** | | **9 (28)** | **26 (336)** | **0.28** | **1.09** | **110.67** |
| *Phase 2* | | | | | | |
| below house | 30 | 5 (16) | 11 (80) | 0.55 | 1.74 | 52.23 |
| chicken shed | 16 | 2 (16) | 1 (80) | 0.04 | 0.12 | 1.42 |
| Pigsty | 9 | 3 (16) | 3 (80) | 0.05 | 0.16 | 0.95 |
| **Total** | | **10 (48)** | **15 (240)** | **0.21** | **0.67** | **54.61** |

* N° quadrants: total number of quadrants sampled

** N° traps: total number of traps set

For the second collection period, below house and chicken shed environments were selected. A total of 84 newly emerged phlebotomines were captured in below house (26% of the total of newly emerged individuals) and chicken shed environments (74% of the total of newly emerged individuals) with a total effort of with 72144 emergence trap-nights. Newly emerged phlebotomines were captured in below house in seven farms (87.5% of total farms) and in chicken shed in four farms (Table 3). In addition to *Ny. whitmani* (92.9% of the total of newly emerged individuals), three individuals of other species were found during this period: one female of *Brumptomyia* sp., and two females of *Pintomyia monticola* (Costa Lima) and *Expapillata firmatoi* (Barretto, Martins & Pellegrino). It was not possible to identify three specimens (3.6%) due to bad conditions of the individuals. Although *Ny. whitmani* was found in both environments, 69% of the individuals of this species emerged in chicken shed. Four farms were the most productive ones, accumulating 94% of the total production of newly emerged phlebotomines (Table 3). The farm sampled in both collection periods (farm 5) showed an abundance of newly emerged individuals in the second period that is comparable to that observed in farms 7 and 8 in below house (Table 3).

A total of 13,993 adult phlebotomines were captured in 147 light traps-night (n = 692, 73 light traps-night corresponding to eight houses; n = 13281, 74 light-traps/night corresponding to eight chicken sheds). Eight species were captured in chicken shed: *Ny. whitmani* (96.28%), *Mg. migonei* (1.47%), *Brumptomyia* sp., (1.05%), *Ex. firmatoi* (0.45%), *Micropygomyia*

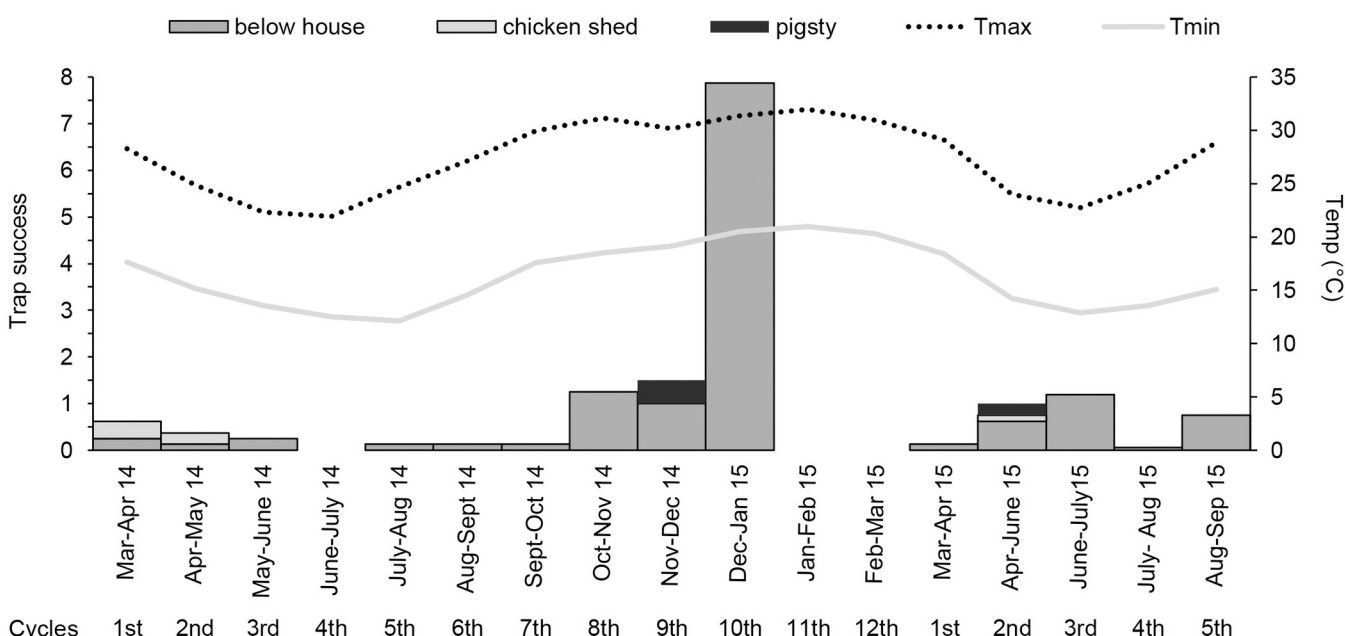

**Fig 2.** Trap success (number of newly emerged individuals/ number of traps) in each cycle during in the two phases of the first collection period: phase 1 (from March 16th 2014 to March 26th 2015–379 nights -12 cycles) and phase 2 (from March 26th 2015 to October 7th 2015–195 nights nights—5 cycles), in each environment: below house (dark gray), chicken shed (light gray) and pigsty (black); average maximum temperature (˚C) (dotted line) and minimum temperature (solid line).

*quinquefer* (Dyar) (0.20%), *Pi. fischeri* (Pinto), *Ev. cortellezzii* and *Py. monticola* (the three species accounting for 0.06%). The latter three species were not captured in the house environment. Regarding the association between abundances of newly emerged and adult

**Table 3.** *Second collection period.* Capture of newly emerged phlebotomines in below house (BH) and chicken shed (CHS) environments in eight farms, from 21st October 2015 to 6th April 2016 (4 cycles, 167 nights): Size of each environment (Size, m²), Number of quadrants with newly emerged phlebotomines (number of sampled soil quadrants in each environment where at least one phlebotomine emerged); Number of traps with newly emerged phlebotomines (total number of traps set in each environment where at least one phlebotomine emerged), trap success (number of newly emerged phlebotomines/number of traps), daily productivity of each environment (number of newly emerged phlebotomines/m²/day) and total period productivity (number of newly emerged phlebotomines on the total area of each environment on each farm for the total collection period).

| Farm | Size (m²) | | N° quadrants with newly emerged phlebotomines (total quadrants) | | N° traps with newly emerged phlebotomines (total traps) | | Trap success | | Daily productivity per square meter | | Total period productivity | |
|---|---|---|---|---|---|---|---|---|---|---|---|---|
| | BH | CHS | BH | CHS | BH | CHS | BH | CHS | BH | CHS | BH | CHS |
| 1 | 50 | 24 | 3 (27) | 6 (42) | 3 (108) | 6 (168) | 0.03 | 0.05 | 0.08 | 0.16 | 681.79 | 631.14 |
| 2 | 24 | 12 | 1 (25) | 0 (24) | 1 (100) | 0 (96) | 0.01 | 0 | 0.03 | 0 | 117.81 | 0 |
| 3 | 12 | 12 | 1 (34) | 0 (19) | 1 (136) | 0 (76) | 0.01 | 0 | 0.02 | 0 | 43.31 | 0 |
| 4 | 32 | 4 | 3 (38) | 0 (21) | 3 (152) | 0 (84) | 0.02 | 0 | 0.06 | 0 | 310.03 | 0 |
| 5* | 30 | 16 | 6 (40) | 4 (24) | 6 (160) | 4 (96) | 0.04 | 0.04 | 0.13 | 0.12 | 644.29 | 245.44 |
| 6 | 24 | 6 | 0 (22) | 0 (19) | 0 (88) | 0 (76) | 0 | 0 | 0 | 0 | 0 | 0 |
| 7 | 64 | 16 | 3 (19) | 8 (22) | 3 (76) | 8 (88) | 0.04 | 0.17 | 0.11 | 0.50 | 1240.14 | 1338.78 |
| 8 | 30 | 12 | 3 (28) | 6 (25) | 3 (112) | 7 (100) | 0.04 | 0.34 | 0.10 | 1.00 | 525.95 | 2002.82 |
| *Subtotal* | 266 | 102 | 20 (233) | 24 (196) | 20 (932) | 25 (784) | 0.03 | 0.08 | 0.07 | 0.23 | 3563.33 | 4218.19 |
| *Total* | 368 | | 44 (429) | | 45 (1716) | | 0.05 | | 0.15 | | 7781.52 | |

*Farm 5 is the farm sampled in the first collection period.

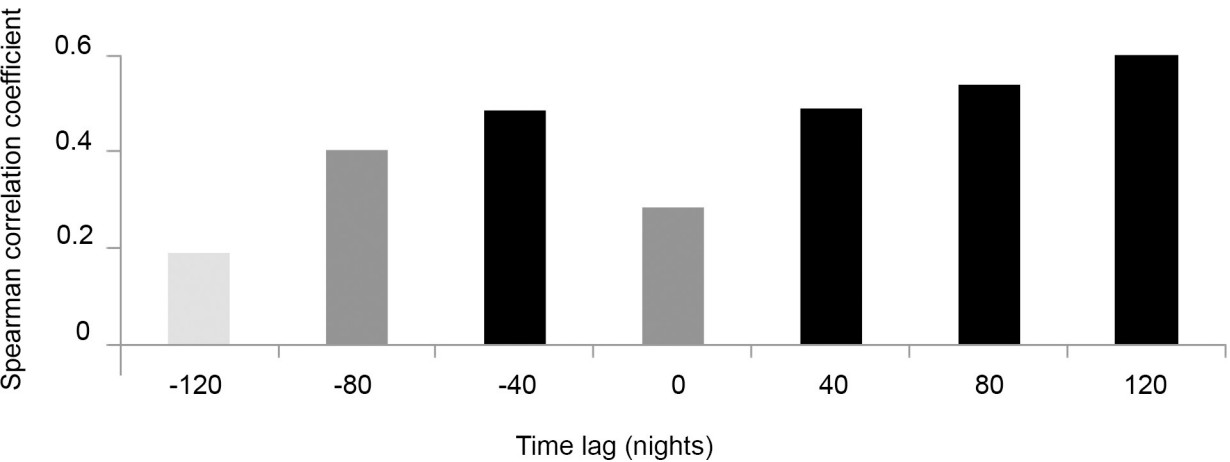

**Fig 3. Spearman correlation coefficient between emergence trap success (number of newly emerged individuals/number of traps) in each of the four biological cycles (≅ 40 nights) and light trap success of adults recorded at different time lags from the same cycle (time 0) up to 3 cycles time lag (120 nights) of the second collection period in both environments.** Different p-values are indicated with different bar colors: black (p <0.01), dark grey (p <0.05) and light grey (p>0.05).

phlebotomines, a positive association was found between trap success of newly emerged phlebotomines and of adults both 40 and 80 nights before and after phlebotomine emergence (Fig 3). A positive correlation was also observed between trap success of newly emerged phlebotomines and of adults observed after 120 nights.

The most parsimonious regression analysis model at the plot level showed that the abundance of newly emerged phlebotomines was explained by the interaction between the environment and the number of chickens, as well as by distance to the forest edge and the cycle period (Table 4). Abundance of newly emerged phlebotomines (trap success) was higher with a higher number of chickens, with chicken shed showing the greatest increase (Fig 4A). In addition, an increase in the abundance of newly emergent phlebotomines was observed in sites close to the forest edges (Table 4 and Fig 4A). The fourth cycle (March-April) showed a lower abundance of newly emerged phlebotomines compared to the first and second cycles (cycle 1: mean: 0.01,

**Table 4. Generalized linear mixed models of the variation of abundance in phlebotomine emergence (a) at the plot level and (b and c) at the quadrant level.** The variables used in the model were environments (chicken shed and below house), number of chickens, distance to the forest edge, cycle (cycle 1, cycle 2, cycle 3, and cycle 4) and farm factor, which was included as random factor. Significance: $p<0.05^*$. Model: Abundance of emergent ~ environment: number of chickens + distance to the forest edge + cycle + random effect farm.

|  | Coefficient | S.D. | p-value |
| --- | --- | --- | --- |
| Intercept | -4.5564 | 0.4648 | <2e-16* |
| chicken shed | 0.2365 | 0.3513 | 0.500786 |
| number of chickens | 1.1526 | 0.5389 | 0.032446* |
| distance to the forest edge | -1.8858 | 0.8088 | 0.019723* |
| cycle 2 | 0.5005 | 0.2739 | 0.067648 |
| cycle 3 | 0.1232 | 0.2965 | 0.677689 |
| cycle 4 | -1.2631 | 0.4593 | 0.005960* |
| chicken shed: number of chickens | 1.3236 | 0.3764 | 0.000437* |
| farm variance: 0.20 |  |  |  |
| Overdispersion factor:1.55 |  |  |  |
| Residual deviance: 164.6 residual Df:55 |  |  |  |

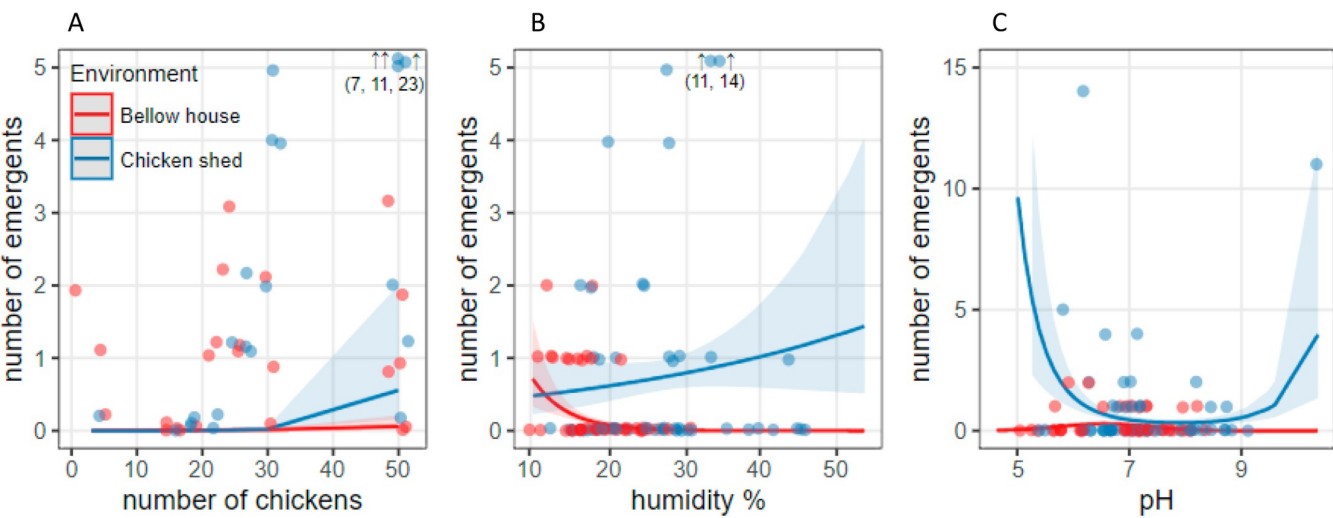

**Fig 4.** Trap success in each environment (number of newly emerged phlebotomines /the number of trap-nights) in each environment (below house and chicken shed) as a function of (A) number of chickens present on the farm, (B) soil humidity percentage (C) soil pH values. Dots: abundance of newly emerged phlebotomines. Lines: values predicted by the corresponding models (see Table 4). Shaded areas are confidence intervals. Abundances observed outside the ranges are indicated with their value and an arrow. Note that some observations indicated with arrows were relocated (observed values are reported in brackets).

SD: 0.014; cycle 2: mean: 0.022, SD:0.057; cycle 4: mean: 0.0039, SD:0.011; Tukey's contrast test, p<0.05), whereas the third cycle (January-February) showed intermediate abundance of newly emerged phlebotomines compared to the other cycles (cycle 3 = mean:0.014, SD: 0.029).

At the quadrant level, 158 soil samples were collected in chicken shed and below house environments from farms 5, 7 and 8. Soils had an average of 23.3% (SD: 5.7) relative humidity, pH 7.1 (SD: 0.35), 10.4% (SD: 5.6) organic matter, 0.7% (SD: 0.4) total nitrogen, 4.7% (SD: 2.5) organic carbon and 9.9 (SD: 1.0) carbon/nitrogen ratio. In the RDA, the variable plot explained 50.9% of the covariability between soil physicochemical variables (F = 31.54, d.f. = 5, p = 0.001) (Fig 5) and was related to environment (36%) and farm separately (7%). The below hou*s*e plots of the three farms and the chicken shed plot of farm 7 were characterized by higher values of carbon/nitrogen and lower values of nitrogen, organic matter, organic carbon, humidity, and pH, compared to chicken shed plots on farms 5 and 8, which showed the opposite characteristics (Fig 5). Two regression models were the most parsimonious at the quadrant level. The first one showed that the abundance of newly emerged phlebotomines depended on the interaction between the environment and soil moisture (Table 5). In the chicken shed environment, the highest number of newly emerged phlebotomines was related to soil substrates with a high percentage of humidity (50%). In the below house environment, although at low abundance, newly emerged phlebotomines were observed in soil substrates with low humidity (10 to 20%, Fig 4B). The second model showed that abundance of newly emerged phlebotomines depended on the interaction between the environment and the pH in a quadratic relation (Table 6). In the chicken shed environment, the number of newly emerged phlebotomines decreased with increasing pH values (between 5 and 7) (Fig 4B), whereas in the below house environment, the abundance of newly emerged phlebotomines did not change with soil pH (Fig 4B).

## Discussion

This is the first report of breeding sites of the phlebotomine species *Ny. whitmani*, which were characterized by sampling newly emerged individuals in an endemic rural area of tegumentary

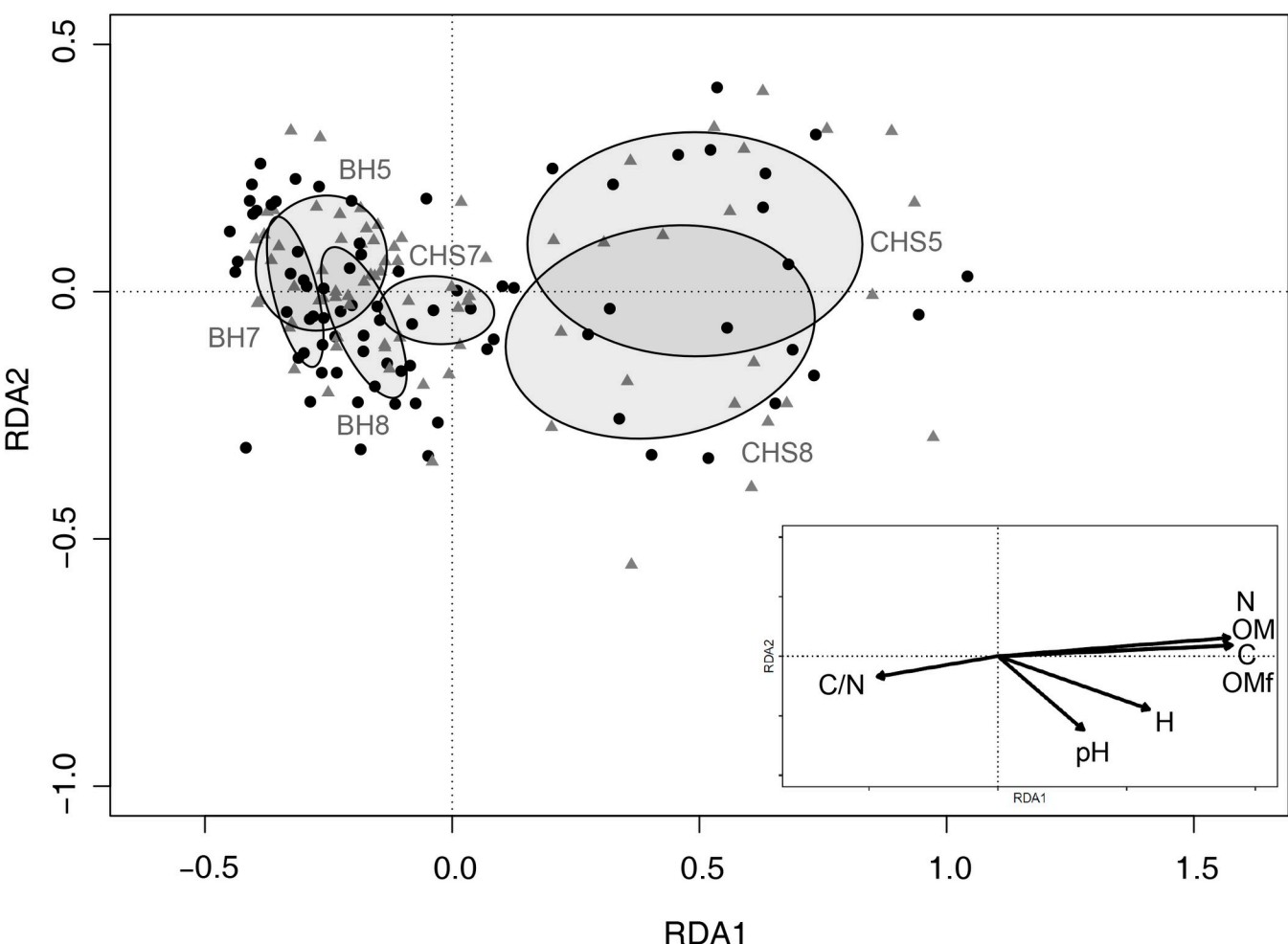

**Fig 5. Redundancy analysis (RDA) of soil plots of the environments below house (BH) and chicken shed (CH) of the sampled farms (5, 7, and 8) as explanatory variables and soil physicochemical variables as response variables: Carbon/nitrogen (C/N), % humidity (H), % organic carbon (C), % total organic matter (OM), % organic matter fractions (OMf), % nitrogen (N) and pH.** Each ellipse shows one of the six plots of the three sampled farms. The variable plot explained 50.9% of the covariability between soil variables.

leishmaniasis; in addition, natural breeding sites of phlebotomine are reported in Argentina for the first time. Breeding sites of this vector species were previously identified through the capture of about 10 individuals [11,20]. In this work, 172 individuals were captured and identified; moreover, the conditions of the breeding sites were characterized, and the collection environments were compared. Emerging individuals of *Ny. whitmani* were collected from

**Table 5. Generalized linear mixed model of the variation of abundance in phlebotomine emergence at the plot level.** The variables used in the model were environments (chicken shed and below house), relative humidity of soil (H) and plot factor was included as random factor. Significance: p<0.05*. Model: abundance of emergent ~ environment: H + random effect plot.

| | Coefficient | D.E. | P-value |
|---|---|---|---|
| intercept | -3.1663 | 0.8432 | 0.000173* |
| chicken shed | 2.7292 | 0.8918 | 0.002210* |
| H | -1.9268 | 0.9636 | 0.045547* |
| environment:H | 2.0872 | 1.0124 | 0.039237* |

Residual Deviance: 221.3 residual df:152

**Table 6. Generalized linear mixed model of the variation of abundance in phlebotomine emergence at the quadrant level.** The variables used in the model were environments (chicken shed and below house), pH and $pH^2$. Significance: $p < 0.05^*$. Model: Abundance of newly emerged individuals ~ environment + pH + environment: pH2 + random effect plot.

| | Coefficient | D.E | P-value |
|---|---|---|---|
| Intercept | -1.4222 | 0.4541 | 0.00174* |
| chicken shed | 0.6192 | 0.5570 | 0.26629 |
| pH | -0.7295 | 0.3321 | 0.02802* |
| $pH^2$ | -0.8245 | 0.4835 | 0.08815 |
| Environment:$pH^2$ | 1.2331 | 0.5470 | 0.02418* |

Residual deviance: 217.8 residual df: 151

domestic and peridomestic environments, where six environments that are considered candidate breeding sites on a farm were evaluated. Sampling was continuous for 574 nights. Once the environments with breeding sites were identified, the search was intensified in the most productive environments on eight farms during the time identified as of maximum emergence for *Ny. whitmani*.

Animal pens were previously indicated as breeding sites of phlebotomine species [12,13,19,36–39]. Here, we found that the main breeding sites of *Ny. whitmani* were the chicken shed, under house and pigsty, which differed from other farm environments, such as forest edge, under fruit tree and experimental chicken shed, where no emerging phlebotomines were recorded. The lack of records of emerging individuals in the latter three environments might be due to the small area sampled. Furthermore, these results suggest that, despite being available in the peridomicile throughout the year, the environments forest edge, under fruit tree and experimental chicken would not be sites more often selected for the development of immature stages of *Ny. whitmani*, since they are more exposed to environmental or human disturbances and regarding the closeness of reproductive adults these sites have few or no blood sources nearby.

Previous studies reported that *Ny. whitmani* adults can be very abundant in the forest edge and primary vegetation sites near the farms [8,22,40]. This high abundance might be due to the fact that adults with source populations in the primary-secondary vegetation are attracted from other nearby environments (e.g. chicken shed, below house, pigsty) by the light source of the traps and are intercepted by traps in the ecotone; however, an increase in abundance due to being a refuge area or an edge effect cannot be discarded [41]. Moreover, in the present study we observed that the environments chicken shed and below house that are closer to the forest edge are the ones with highest productivity of newly emerged individuals; thus, they might be the first colonization sites from the source population [42]. The proximity of chicken sheds and houses to the forest edge would also involve the availability of vegetable sources of carbo hydrates in the forest, along with blood sources in the domestic and peridomestic environments.

The peridomestic environments provide phlebotomine adults with refuge, especially the chicken sheds, because they reduce the variation of outside temperature, are a source of attractant allomones and provide a concentrated blood source [43]. In turn, chicken feces are attractive to females for oviposition site selection [37]. While chicken sheds were the main producers of emerging individuals in this study, soil organic matter did not explain the presence of newly emerged individuals; rather, the variable environment was the one explaining those differences, as shown in the redundancy analysis. These results show the importance of multidimensional design of experiments as a tool to understand the biological processes in cases in which the effects of environment cannot be separated from the soil components. In

this sense, we do not discard the importance of the availability of organic matter in the soil of chicken sheds for the development of the immature stages of *Ny. whitmani.*

The soil surrounding houses was identified as a phlebotomine breeding site in villages in India [38], where houses are built close to cattle sheds. However, the importance of houses as breeding site has never been reported for the species of Phlebotominae from our region. In the study area, several of the rural houses are built on wood stilts raised from the ground. Elevation avoids possible floods and access of poisonous animals to the indoor area (serpents and arthropods). The space below houses is frequently used by domestic, farmyard and synanthropic animals as resting sites, where they find protection from the sun and a cool environment; in these sites, unused objects and organic matter are usually accumulated (Manteca-Acosta, pers. obs.). The characteristics of these sites under the house supported the hypothesis that the soil below the house might be an optimum habitat for development of the phlebotomine larval stages. Our results support this hypothesis, showing that the environment under the house was the one with second highest abundance of emerging individuals; in addition, below house and chicken shed exhibited similar values in terms of total contribution to the population (productivity). While the chicken shed soil is more productive per square meter, the below house environment produced a higher number of newly emerged individuals in most of the farms because this environment is larger than the chicken sheds (it covers up to eight times the area of a chicken shed). In this sense, this environment of the study area would be a site with optimum characteristics for the breeding of *Ny. whitmani*; indeed, although below house was not the main source breeding site, it contributed to productivity generated by the chicken shed in the peridomicile, increasing the risk of vector-human contact. Once emerged from under the house, phlebotomines can enter the intradomicile through the openings in the wooden floor, attracted by people that are sleeping in the house. These results show the need to consider interventions not only in animal sheds, as indicated in other studies, but also, from the relative risk perspective, in specific areas of the peridomicile environments according to local cultural practices that modulates population dynamics of vectors and potential reservoirs.

On the other hand, capture using emergence traps modified from Casanova et al. (2013) was effective in the recovery of newly emerged individuals of *Ny. whitmani*, as well as, in lower abundance, of the species *Brumptomyia* sp., *Py. monticola* and *Ex. firmantoi*. Likewise, considering the study period of the present work, the total sampling area and the number of traps set, the estimated total daily production of *Ny. whitmani* in soils of chicken sheds and under houses was a little lower than that reported for the species *Lu. longipalpis* in chicken shed environments in the locality of Promissao, Brazil (0.41 emerging individuals per $m^2$ per day, Casanova et al., 2013) and higher than the value recorded for other phlebotomine species typical of wild environments of Brazilian forests (0.04 emerging sand flies per $m^2$ per day, Arias & De Freitas, 1982) [44]. However, the recorded productivity was not homogeneous over the study period or across space, either between soil plots of a single environment in the different farms or within plots of the same farm. The exhaustive sampling involving 2292 emergence traps in both collection periods, the concentration of newly emerged individuals in a few traps and the high number of traps with no newly emerged individuals evidence a contagious distribution pattern in *Ny. whitmani* towards specific soil quadrants, as observed in other vector phlebotomine species [12,13]. The contagious spatial distribution is the pattern commonly reported for the insects in natural environments [45], and has been suggested for phlebotomine species in the forest soil [36]. Thus, at the microscale, females would have oviposition hotspots, i.e. microsites that might be identified by microbiological and physicochemical characteristics to support environmental management interventions or chemical control. According to our results, soil moisture and pH might be environmental cues for females of this species; however,

alternatively, those cues might be affecting development and survival of immature stages, a topic that should be further explored.

In the present work, the number of adults captured with light traps was several orders of magnitude higher than the number of newly emerged individuals captured in the same environments (166.6 adults captured in light traps per emerging individual captured in emergence traps), which is consistent with previous works [12,13,15]. These disparities between catches of adults and newly emerged might be attributed to differences in the sampling method as the emergence trap collects only teneral insects inside the trap area while the light trap attracts flying adults from farther away, not only by the luminous bait but also due to nearby blood sources [12]. This difference is maintained despite greater capture effort with emergence traps than with light trap (487 emergence traps per light trap-night for the second collection period). Furthermore, the mean lifespan of *Ny. whitmani* adults in a natural environment and, therefore, how these populations will be structured according to their life cycle stages are also unknown factors; therefore, individuals of multiple emerging generations might accumulate in a light trap due to overlap of cohorts or to climatic conditions that generate optimum "windows" during adverse periods. However, the comparison of the number of newly emerged individuals produced in each environment, rather than the number of individuals captured in the emergence traps, yields a similar size of both immature and adult subpopulations. Thus, the chicken sheds of the present study would have a production of newly emerged individuals that represents 30% of the adults captured with light traps in that environment, whereas below house, that proportion would be higher, representing 500% of the number of adults captured in this environment. These results suggest movement of individuals between those environments within a farm, which would belong to the same population, with adults captured in an environment possibly having bred in the other. In turn, these results support the evidence to consider under house and chicken shed as the most important breeding sites in the farms of the study area. Likewise, the temporal association of immature and adult abundances showed that abundances are associated with the known development time of the immature stage. This explains that a higher abundance of adults will produce a higher number of newly emerged individuals after some weeks and, in turn, the increase of immature individuals will further increase the total number of adults. While there are records of the mean lifespan of the *Ny. whitmani* adult in the laboratory (at least 20 days, Moraes Ribeiro et al. 2015) [46], this value is still unknown for natural environments. Based on these results and on the observed association, we may assume that this period would be of at least 120 nights. If the life span were longer, no association between the number of adults and the number of newly emerged individuals would be expected with these time lags, since the number of adults would be the result of the accumulation of numerous generations of newly emerged individuals rather than of the immediately previous emergence. Similarly, we may interpret that, since the increase of adults produces an increase in the number of newly emerged individuals after a short period (Fig 4), there would be a short latency period for this species in this area, at least during the spring and summer months.

Finally, the temporal monitoring of newly emerged individuals for more than one year allowed us to explain the seasonal variation patterns of adults, which was previously reported in the study area [8,22,40]. Indeed, the warmest months are those of greatest emergencies, and this is consistent with the moments of the year with maximum adult abundances.

To conclude, the results suggest that management of number the chicken in the sheds, and the decision of where to localize the chicken sheds in relation to the houses and the forest edge might contribute to the decreased of *Ny. whitmani* abundance on farms. Moreover, below house was found to be as important as chicken sheds as breeding sites of *Ny. whitmani*. Experimental studies are necessary to evaluate the effect of the application of environmental

management practices for the prevention and control of this vector, and their acceptance by rural people, associated with our results, like the management of soil moisture and pH, the number of chickens and the localized the chicken sheds.

## Acknowledgments

To all the neighbors who opened their farms to us to carry out this study. Also, to the entire teamwork of National Institute of Tropical Medicine, especially to Nilso Molina, Juan Molina, Micaela Fanucce and Marcio Antunez.

## Author Contributions

**Conceptualization:** Oscar Daniel Salomón.

**Data curation:** Mariana Manteca-Acosta, Regino Cavia, María Eugenia Utgés.

**Formal analysis:** Mariana Manteca-Acosta, Regino Cavia, María Eugenia Utgés.

**Funding acquisition:** Oscar Daniel Salomón, María Soledad Santini.

**Investigation:** Mariana Manteca-Acosta, María Eugenia Utgés, Oscar Daniel Salomón, María Soledad Santini.

**Methodology:** Mariana Manteca-Acosta, Oscar Daniel Salomón.

**Project administration:** Oscar Daniel Salomón, María Soledad Santini.

**Resources:** Oscar Daniel Salomón.

**Supervision:** Oscar Daniel Salomón, María Soledad Santini.

**Validation:** Mariana Manteca-Acosta, Regino Cavia.

**Visualization:** Mariana Manteca-Acosta, Regino Cavia, Oscar Daniel Salomón, María Soledad Santini.

**Writing – original draft:** Mariana Manteca-Acosta, Regino Cavia, Oscar Daniel Salomón, María Soledad Santini.

**Writing – review & editing:** Mariana Manteca-Acosta, Regino Cavia, María Eugenia Utgés, Oscar Daniel Salomón, María Soledad Santini.

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
