## [Decision Letter · Decision Letter 0]

3 Jun 2021

Dear Dra. Manteca-Acosta,

Thank you very much for submitting your manuscript "Peridomestic natural breeding sites of Nyssomyia whitmani (Antunes and Coutinho) in an endemic area of cutaneous leishmaniasis in northeastern Argentina" for consideration at PLOS Neglected Tropical Diseases. As with all papers reviewed by the journal, your manuscript was reviewed by members of the editorial board and by several independent reviewers. The reviewers appreciated the attention to an important topic. Based on the reviews, we are likely to accept this manuscript for publication, providing that you modify the manuscript according to the review recommendations. 

Sincerely,

Paul Andrew Bates

Associate Editor

Jesus Valenzuela

Deputy Editor

Reviewer's Responses to Questions

**Key Review Criteria Required for Acceptance?**

**Methods**

-Are the objectives of the study clearly articulated with a clear testable hypothesis stated?

-Is the study design appropriate to address the stated objectives?

-Is the population clearly described and appropriate for the hypothesis being tested?

-Is the sample size sufficient to ensure adequate power to address the hypothesis being tested?

-Were correct statistical analysis used to support conclusions?

-Are there concerns about ethical or regulatory requirements being met?

Reviewer #1: (No Response)

Reviewer #2: The objetives are clear and the study design is appropiate to resolve them.

Sample size and statistical analysis are correct and the entire procedure was explained in detail.

Reviewer #3: Diseases for review is a Peridomestic natural breeding sites of Nyssomyia whitmani (Antunes and Coutinho) in an endemic area of cutaneous leishmaniasis in northeastern Argentina. Breeding sites, environmental and structural characteristics were used to identify adult abundance. It is interesting as it focuses on the direct field associated with Phlebotominae vectors breeding sites (and little is known on this matter). I have the following comments.

Specific comments.

Lines 114 to 115. ‘in the warmest month (January), and 7°C and 15°C in the coldest month (July), respectively” how the 15c coldest month? How about the humidity of the region?

Line 120. Each of the seasons was used the same number of traps was used. 

Line 128. Please add the manufacturer's detail PVC pipe tube.

Line 134. If trap was checked every 15-20 nights, how author would say “newly recorded as a newly emerged individual”. Also, as the author suggested keeping 15 -20 might cause deterioration of the fly, does this itself might cause repellent or attract the fly? Any prediction? 

Line 147 to 149. Was this done at the same locations?

Lines 187 to 190. I suggest using the COI amplification and sequencing methods to identify the species of the fly. 

The authors stated that “physicochemical characteristics (Table 1)” I don’t see any data on the table. 

References need any additional additions, particularly many of the scientific names were not italicized.

**Results**

-Does the analysis presented match the analysis plan?

-Are the results clearly and completely presented?

-Are the figures (Tables, Images) of sufficient quality for clarity?

Reviewer #1: The article presents an important contribution to the knowledge of breeding sites of Nyssomyia whitmani, a species of sandfly vector of L. braziliensis, etiological agent of tegumentary leishmaniasis (TL) in Argentina and Brazil. Worldwide, the identification of breeding sites for leishmaniasis vectors represents a major gap in the knowledge of the biology and ecology of sand flies. Certainly, the results of this study are of great epidemiological importance, because, within an integrated control strategy, it reinforces a real perspective of changing the vector control paradigm to be directed exclusively to the adult population. The introduction and justification of the work proposal is duly endorsed and updated, highlighting the importance of the study to improve the strategies for monitoring and controlling TL. The methodology was clearly presented and involved a huge collection effort, which was facilitated by the authors' proposal to develop the study in periods and phases, in order to implant a larger sample in the preferred breeding sites discovered in the first phase of the study. The results were presented and analyzed clearly and presented an adequate number of figures and self-explanatory tables. Here are some suggestions and minor corrections:

- In the item Emergence trap: I am curious to know why only one trap size was used. I believe that, using the same collection effort, larger traps (with a larger diameter) would increase the sampled area, ok? If possible, I think it would be interesting to present an image of the emergency traps used and, preferably, fixed in any of the investigated environments. 

- In the last line of page 5 - I believe that the collection effort is better defined as the number of traps and-or the sampled area, regardless of whether or not new emerged sand flies are captured.

-In Results: “The period of greatest abundance of newly emerged phlebotomines was from October to January 2014, with a…” add the years: October 2014 to January 2015, ok?

- Tale 1 and 2, draws attention to the difference in the values of “Total period productivity” for Chicken shed from Farm 5, as both were sampled in both periods and the capture of emerged sand flies was practically the same with the use of a same number of emergency traps (96). Review, okay? In addition, I believe that it would facilitate the understanding of these estimates to include information on the size (m2) of each environment in each Farm.

- In Discussion: In Brazil it is very common for chickens to be loose in the peridomicile, where they sleep on top of trees. Do you have this situation in any farm? and did you get to sample any soil under these trees?

- Have other insects, possibly predators of the immature forms of sand flies, been collected in the emergence traps?

- regarding possible associations between the newly emerged in emergence trap and adults captured with light traps population, I would like to raise some questions for discussion with the authors; is there any information on latency (or diapause) of any phase of the sand flies life cycle for sub-tropical and tropical regions? Is it possible for males and females to survive for more than 30 days? Marking-release-recapture studies do not show this, okay? Possible associations between abundance of emerged in emergence trap and adults captured with light traps with a time difference greater than 2 months, it seems to me to be more influenced by climatic and sampling factors, on both populations, which are difficult to count. it is likely that the climatic conditions on the days of the collections, or on the previous days, influence more collections of adult forms than immature forms.

- an issue that always leaves a doubt for researchers trying to discover the preferred breeding sites of sand flies is the uncertainty about whether the number of samples in each investigated environment was sufficient to determine their favorable or unfavorable condition to serve as a breeding site. Not to mention the existence of other possible (and unknown) sites. Did the authors also get this feeling?

Reviewer #2: The results are presented in a clear and easy-to-read manner despite the amount of information contained in the research work. The number and format of tables and graphs are sufficient and adequate.

Reviewer #3: (No Response)

**Conclusions**

-Are the conclusions supported by the data presented?

-Are the limitations of analysis clearly described?

-Do the authors discuss how these data can be helpful to advance our understanding of the topic under study?

-Is public health relevance addressed?

Reviewer #1: Yes.

Reviewer #2: The conclusions are very well supported by the available dataset and also discussed limitations with proposals for future work. 

As a general product, it has concrete and valuable information to implement a control design at the breeding site level that will serve as a complement for other strategies to be used.

Reviewer #3: (No Response)

**Editorial and Data Presentation Modifications?**

Reviewer #1: (No Response)

Reviewer #2: The manuscript is attached with some minor corrections inside.

Reviewer #3: (No Response)

**Summary and General Comments**

Reviewer #1: The authors are to be congratulated for the undertaking of enormous fieldwork for a year and a half.

Reviewer #2: The research work presented represents a great contribution to the study of the ecology of the immature stages of the subfamily Phlebotominae. There are very few studies on the search for breeding sites precisely because it represents a great challenge not only in the sampling design to optimize the capture, but also of all the field work and especially laboratory work to be able to observe that large amount of material with so little chance of success.

Reviewer #3: The paper is clear, relatively well written and the study is well designed and organized. It merits publication, in my opinion, after some changes and revision (see below for specific comments).

PLOS authors have the option to publish the peer review history of their article (what does this mean?). If published, this will include your full peer review and any attached files.

Reviewer #1: Yes: Claudio Casanova

Reviewer #2: No

Reviewer #3: No

Figure Files:

Data Requirements:

Reproducibility:

References

---

## [Editor Report · Decision Letter 1]

22 Jul 2021

Dear Dra. Manteca-Acosta,

We are pleased to inform you that your manuscript 'Peridomestic natural breeding sites of Nyssomyia whitmani (Antunes and Coutinho) in an endemic area of tegumentary leishmaniasis in northeastern Argentina' has been provisionally accepted for publication in PLOS Neglected Tropical Diseases.

Best regards,

Paul Andrew Bates

Associate Editor

Jesus Valenzuela

Deputy Editor

---

## [Editor Report · Acceptance letter]

5 Aug 2021

Dear Dra. Manteca-Acosta,

We are delighted to inform you that your manuscript, "Peridomestic natural breeding sites of Nyssomyia whitmani (Antunes and Coutinho) in an endemic area of tegumentary leishmaniasis in northeastern Argentina," has been formally accepted for publication in PLOS Neglected Tropical Diseases.

Best regards,

Shaden Kamhawi

co-Editor-in-Chief

Paul Brindley

co-Editor-in-Chief
